# Application of the Artificial Neural Network to Predict the Bending Strength of the Engineered Laminated Wood Produced Using the Hydrolyzed Soy Protein-Melamine Urea Formaldehyde Copolymer Adhesive

**Morteza Nazerian [1]** , **Fatemeh Naderi [1]** and **Antonios N. Papadopoulos [2],***

1   Department of Bio Systems, Faculty of New Technologies and Aerospace Engineering, Shahid Beheshti University, Tehran 1983969411, Iran; m_nazerian@sbu.ac.ir (M.N.); naderi1393f@gmail.com (F.N.)
2   Laboratory of Wood Chemistry and Technology, Department of Forestry and Natural Environment, International Hellenic University, GR-661 00 Drama, Greece
*   Correspondence: antpap@for.ihu.gr

**Abstract:** The artificial neural network (ANN) was used to predict the modulus of rupture (MOR) of the laminated wood products adhered by melamine/urea formaldehyde (MUF) resin with different formaldehyde to melamine/urea molar ratios combined with different weight ratios of the protein adhesive resulting from the alkaline treatment (NaOH) of the soybean oil meal to MUF resin pressed at different temperatures according to the central composite design (CCD). After making the boards and performing the mechanical test to measure the MOR, based on experimental data, different statistics such as determination coefficient ($R^2$), root mean square error (RMSE), mean absolute error (MAE) and sum of squares error (SSE) were determined, and then the suitable algorithm was selected to determine the estimated values. After comparing estimated values with the experimental values, the direct and interactive effects of the independent variables on MOR were determined. The results indicated that using suitable algorithms to train the ANN well, a very good estimate of the bending strength of the laminated wood products can be offered with the least error. In addition, based on the estimated and measured strengths and FTIR and TGA diagnostic analyses, it was found that the replacement of the MUF resin by the protein bio-based adhesive when using low F to M/U molar ratios, the MOR is maximized if a high range of temperature is used during the press.

**Keywords:** wood laminated product; MUF-modified protein adhesive; optimization; MOR; ANN

## 1. Introduction

Soy flour is the waste of the process of extracting soy bean oil. When extracting oil, about 4.5 tons of waste flour is produced per each ton of oil [1] and using it can be very useful both economically and environmentally. So far, formaldehyde-based synthetic resins have been the main adhesive used to produce wood products while in addition to limitations in using fossil resources, the release of formaldehyde gas has made their limitations more evident [2,3]. Hence, the adhesives resulting from biomass waste resources can be a suitable alternative to mitigate the negative effects on the environment and avoid dependency on fossil fuel resources. Despite a lot of soy wastes used to feed the livestock, a lot of them are still produced every year [4]. Recently, as the awareness of the environmental effect and resource stability increases, attempts are made to use the soy flour and other biomass resources to make bio-based formaldehyde-free adhesives to product wood products [5–9]. To overcome the disadvantages of using the bio-based materials such as dry and wet strengths of their products, activities are performed by applying unsaturation process such as acid treatments [10] and alkaline treatment [11] to denature the protein together with using the petrochemical crosslinkers and bio-based additives

resulting from the wastes of different wood industries. During the denaturation, the protein chain is uncoiled and both hydrophilic and hydrophobic exposed groups are released so that the interaction between the protein molecules and the protein with substrate increases during the curing [12]. In alkaline conditions, the moisture and mechanical properties of the panels adhered by the protein adhesive improve so that the requirements of the strength are met for internal consumptions. Denaturation along with crosslinkers can engage amino-groups on the protein and improve the strength due to ester connections [13,14]. However, using the bio-based additives such as lignin as the crosslinker also usually requires very complicated modification techniques and increases the cost so that using petrochemical materials as the crosslink accelerating agent becomes more cost-effective and it mitigates the environmental issues such as the release of formaldehyde gas. For this purpose, treating the protein chemically and activating the active groups connectable with amine-groups on the molecule of petrochemical materials such as urea, melamine, ect, the bio-based adhesive can be produced.

To optimize the process of producing wood laminated products, all parameters and variables must be considered and tested. However, the artificial neural network modeling can be used to determine the optimal process parameters to make the laminated product without spending a lot of cost, energy and time.

ANNs are an advanced data modeling means used to model the complex undefined nonlinear relations between the inputs and outputs without any preliminary assumptions or mathematical relations existing between them [15]. They are inspired from the biological neural network computational concept. Today, ANNs are applied as one of the most attractive branches of the artificial intelligence in many scientific and engineering applications including prediction, optimization, classification, identification of patterns and data processing [16].

In wood science, the latest studies have shown the capability of ANN to predict the physical and mechanical properties of wood [17–24], wood composite products [25–31] and wood machining [32–34].

To train the relationship between the input and output variables, the neural network is trained together with the data related to the problem being examined through a training algorithm. Training is the process of adjusting the connection weights directing the ANN to produce outputs equal or close to the target values [35]. Meanwhile, the feed forward and back propagation is the most famous training algorithm to train the ANN [36]. The multilayer perceptron (MLP) feed forward neural network is the commonest prediction architecture [22,37]. The MLP architecture includes one input layer, one output layer and one or more hidden layers depending on the complexity of the problem being examined. Each layer in MLP is composed of some interconnected elements known as "neurons" and the neurons direct the network toward producing one special output of the input variables. An example of the MLP architecture is given in Figure 1 based on which the optimized architecture is established.

Based on the Equation (1), the output of the MLP given in Figure 1 is also computed:

$$Y = g\left(\theta + \sum_{j=1}^{m} v_j \left[\sum_{i=1}^{n} f\left(w_{ij}x_i + \beta_j\right)\right]\right) \tag{1}$$

where $Y$ is the predicted value of the dependent variable; $x_i$ is the input of the $i$-th independent variable; $w_{ij}$ is the connection weight of the $n$-th input neuron and the $i$-th hidden neuron; $\beta_i$ is the bias value of the $i$-th hidden neuron; $v_j$ is the connection weight between the $j$-th hidden neuron; $\theta$ is the output bias value; and $f(.)$ are the activation functions of the output and hidden neurons, respectively.

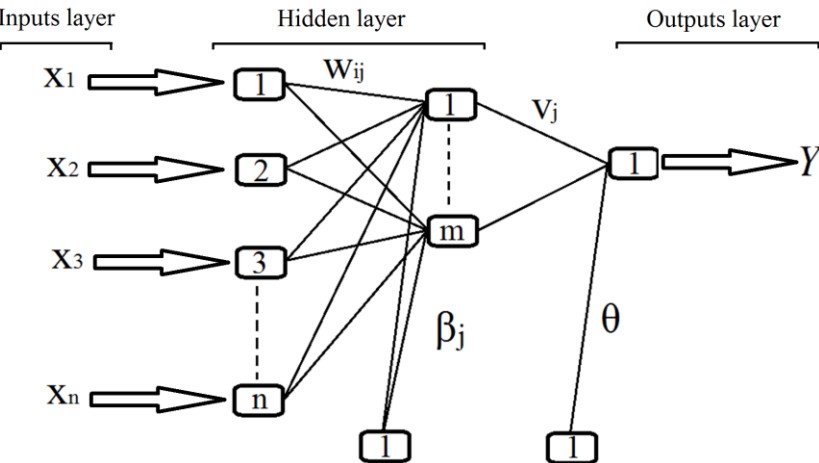

**Figure 1.** A typical multi-layered ANN architecture based on MLP.

In the ANN architecture, the first and last layers are the input and output layers, respectively. The layer between the input and output layers is known as the "hidden layer". The hidden layer receives the data from the input layer and processes the data subsequently and then, it sends them as the response to the output layer. The output layer receives the resulting response from the hidden layer and produces the output data for the network input layer. In this way, it sends the output data to the outside environment [16].

In the relevant literature, the effects of various process variables are discussed on the bending strength of the laminated products in detail. Furthermore, ANN attempts are made to predict the strength properties of the laminated wood products. It is determined that there is very little information on the possibility of using the ANN to estimate the strength properties of the laminated wood products based on the protein bio-based adhesives. Hence, the main purpose of the present research is to examine and predict the bending strength of the laminated products adhered by the plant protein adhesive resulting from the soybean oil meal combined with the melamine-urea formaldehyde (MUF) resin at different molar ratios cured at different press temperatures.

## 2. Materials and Methods

### 2.1. Materials

Edible soybean oil meal (produced by Khoshpak Products Co., Tehran, Iran) containing 53.3 g protein per 100 g) was used after treating it with NaOH (99%), ethylene glycol (density 1.11 kg/L) and HCl (20%) to produce the bio-based protein adhesive. Powder melamine (with the purity 99.8%), urea (purity 46%), formalin (density 1.08 gr/cm$^3$, pH 2.5–4 and concentration 38%), NaOH (40%), butanol and ammonium chloride (20%) were also used to produce melamine urea formaldehyde (MUF) resin.

To make the laminated product and to examine the effect of the dependent variables on the bending strength accurately, samples were prepared randomly from walnut wood (*Juglans regia* L.) grown in the north of Iran with the average age 30 years old and average density 700 g/cm3 after examining some wood species due to the prevention of the internal swelling of the board due to the accumulation of the water vapor under the press. A special emphasis was made to prepare boards from flawless logs. Boards with the dimensions 350 × 70 × 37 mm were prepared radially. After storing them in the laboratory for 3 months and reaching the equilibrium moisture content 15%, the samples were exposed to the temperature 110 °C until the moisture content reached 8%.

### 2.2. Methods

2.2.1. Experimental Design

To determine the effect of the independent variables on the response (MOR) and predict the direct, interactive and square effects of the variables on the response in the

process of using the ANN, the experiments were designed first using the response surface methodology (RSM) and based on the center composite design (CCD). In this process, three distinct points including the axial, factorial and central points were distinguished in a cubic matrix environment and the number of iterations was set at 2 per each point. The levels of each variable including the molar ratio of formaldehyde to melamine/urea (MR), weight ratio of the modified protein (MP) to MUF resin (WR) and the press temperature (Tem) were 1.68:1, 1.805:1, 1.93:1, 20:80, 40:60, 60:40 and 140, 160 and 180 °C, respectively. According to the number of the axial, factorial and central points and 2 iterations for each of these points and based on the equation 2n + (n × 2) + k where n is the number of variables and k is the number of iterations at the cube center,34 test samples were prepared for the bending test.

### 2.2.2. Preparation of the Soybean Oil Meal-Modified Protein Adhesive

Preparation of protein adhesive was done based on previous research [38]. For this purpose, to the aqueous solution (333.33 mL) containing 1.766 g ethylene glycol (as the phase transferrer) and 9.33 alkali (NaOH 99%), 116.66 g soybean oil meal was added that had passed through the sieve with the mesh size 100 after the aqueous solution was heated and its temperature reached 70 °C in a three-necked flask equipped with the mechanical stirrer, thermometer and refrigerator while the mixer rotated with the speed 650 rpm. As the mixture's temperature increased to 88–90 °C for 15 min, the solution was kept at this temperature for 2 h to complete the reaction. Then using the ice bath, the solution's temperature decreased to 35 °C quickly. Using HCl 20%, the mixture's pH decreased from 12.9 to 7 and after passing through the sieve with the mesh size 35 and leach it and remove the lumps, it was kept in the refrigerator at the temperature 3 °C to be used later.

### 2.2.3. Making the MUF Resin and Wood Laminated Product

After installing the three-necked flask equipped with the pH-meter, condenser connected to the water flow and alcohol thermometer in a hot oil bath, all formaldehyde (as formalin solution 38%) required to make each resin with different F:MU molar ratios was loaded in the flask (according to Table 1 as the design of experiment, with different values including 202.7 g equivalent to 1.68 mol, 218.9 g equivalent to 1.805 mol and 235.12 g equivalent to 1.93 mol) and in the first making stage (alkaline stage), the first part of urea equivalent to 79.48 g (or 92%) was added to the formalin solution. Then, the pH of the resulting solution (5.3–5.5) was increased to 8–8.4% by adding NaOH 40%. As the reaction's temperature increased to 55–66 °C and adding butanol in the amount of 2.5% of the total weight of melamine + urea to the solution and all melamine (10% of total urea + melamine equivalent to 9.08 g), the solution was heated for 30 min. In the second (acidic) stage, and controlling the pH by adding ammonium chloride 20%, the pH decreased to 5–5.5. As the solution's temperature increased to 80–85 °C and after keeping it at this temperature for 5–10 min and ensuring the complete solution of urea and melamine through testing the formation of a transparent solution in a cold water medium by adding 2–3 drops of the solution, the methylolation process formed. As the solution's temperature decreased to 60 °C, the remaining component of urea from each of the three molar ratios of F to MU (equivalent to 6.9 g) was also added while the mixer was mixing the solution in all stages. As the heater was turned off and was removed from the oil bath, the solution was put aside to cool down. Finally, adding 2–3 drops of ammonium chloride 20%, the solution's pH became neutral to be more durable. Hence, three types of MUF resin were made with different F to MU molar ratios.

**Table 1.** Design of experiment (actual and coded values of the input factors).

| № | $x_1$ | $x_2$ | $x_3$ | MR | WR | Tem | № | $x_1$ | $x_2$ | $x_3$ | MR | WR | Tem |
|---|---|---|---|---|---|---|---|---|---|---|---|---|---|
| 1 | 1 | 1 | 1 | 1.93 | 60 | 180 | 18 | 1 | −1 | 1 | 1.93 | 20 | 180 |
| 2 | 1 | 1 | −1 | 1.93 | 60 | 140 | 19 | −1 | 0 | 0 | 1.68 | 40 | 160 |
| 3 | 0 | 0 | 0 | 1.805 | 40 | 160 | 20 | −1 | −1 | 1 | 1.68 | 20 | 180 |
| 4 | 0 | 0 | 0 | 1.805 | 40 | 160 | 21 | −1 | 1 | 1 | 1.68 | 60 | 180 |
| 5 | 0 | −1 | 0 | 1.805 | 20 | 160 | 22 | −1 | −1 | 1 | 1.68 | 20 | 180 |
| 6 | 0 | 0 | 1 | 1.805 | 40 | 180 | 23 | 1 | 1 | −1 | 1.93 | 60 | 140 |
| 7 | 1 | −1 | −1 | 1.93 | 20 | 140 | 24 | −1 | 1 | −1 | 1.68 | 60 | 140 |
| 8 | 0 | 1 | 0 | 1.805 | 60 | 160 | 25 | 0 | 0 | 0 | 1.805 | 40 | 160 |
| 9 | −1 | −1 | −1 | 1.68 | 20 | 140 | 26 | 0 | −1 | 0 | 1.805 | 20 | 160 |
| 10 | −1 | 1 | 1 | 1.68 | 60 | 180 | 27 | 1 | 0 | 0 | 1.93 | 40 | 160 |
| 11 | 0 | 0 | 1 | 1.805 | 40 | 180 | 28 | 1 | −1 | −1 | 1.93 | 20 | 140 |
| 12 | 0 | 0 | 0 | 1.805 | 40 | 160 | 29 | 1 | 0 | 0 | 1.93 | 40 | 160 |
| 13 | 0 | 0 | −1 | 1.805 | 40 | 140 | 30 | 0 | 1 | 0 | 1.805 | 60 | 160 |
| 14 | 0 | 0 | −1 | 1.805 | 40 | 140 | 31 | 1 | 1 | 1 | 1.93 | 60 | 180 |
| 15 | 1 | −1 | 1 | 1.93 | 20 | 180 | 32 | −1 | 1 | −1 | 1.68 | 60 | 140 |
| 16 | 0 | 0 | 0 | 1.805 | 40 | 160 | 33 | −1 | 0 | 0 | 1.68 | 40 | 160 |
| 17 | −1 | −1 | −1 | 1.68 | 20 | 140 | 34 | 0 | 0 | 0 | 1.805 | 40 | 160 |

Based on the design of experiment (DOE), the MUF-MP adhesive was obtained with MUF to MP weight ratios equal to 20:80, 40:60 and 60:40 after mixing and putting it in a mechanical stirrer for 15 min with the number of rotations 2000 rpm, and a homogeneous solution was obtained. After the uniform distribution of the adhesive on the upper and lower surfaces of the middle layer and the establishment of the upper and lower layers, the assembly was put into the press under a pressure equal to 30 kg/cm$^2$ for 20 min at a certain temperature. Then, after removing the wood laminated panel from the press, it was air-conditioned for 2 weeks. Then, the panels were cut with the dimensions $350 \times 20 \times 20$ mm and three-point bending test was performed according to the EN 302 Standard with the loading perpendicular to the glue line at the speed 5 mm/s. Equation (2) was used to determine the bending strength of the panel. The results obtained from the bending test of the samples were used to develop a modeling by the ANN approach to predict the bending behavior of panel.

$$MOR = \frac{3Pmax \times L}{2bS^2} \tag{2}$$

where *Pmax* is maximum load (N), *L* is span of panel (mm), *b* is width of panel, and *S* is the panel thickness.

After performing the bending test, the experimental values of the MOR obtained were compared with the results predicted by applying the ANN approach to evaluate the performance of the model developed and determine the effects of the variables, their interactive effects and the optimum level of the application of each variable to produce wood laminated panels, subsequently.

### 2.2.4. Artificial Neural Network (ANN) Analysis

ANN was used both for predicting the bending strength and determining the optimal manufacturing parameters offering the high bending strength of panel. The F to MU molar ratio (MR), the weight ratio of the modified protein (MP) to MUF resin (WR) and the press temperature (Tem) have been the independent variables as inputs in the ANN modeling. The data obtained from the experimental studies were modeled using MATLAB Neural Network Toolbox. The experimental data were categorized into three training, testing and validation data sets to determine the effect of the making process parameters on panel's MOR. The training data set was used to develop the network and the testing data set was used to evaluate the model's performance while the validation data set was used to validate the developed model. To model the MOR, 5 data (15% of all data) were considered for the testing set, 5 data (15% of all data) were considered for the validation set and 24 data (70%

of all data) were considered for the training set. The data sets used for the model prediction, the results of the ANN analysis and the experimental results are given in Table 2.

**Table 2.** The experimental and estimated results of the bending strength test of panels.

| № | $x_1$ | $x_2$ | $x_3$ | Actual Value | Predicted Value | № | $x_1$ | $x_2$ | $x_3$ | Actual Value | Predicted Value |
|---|---|---|---|---|---|---|---|---|---|---|---|
| 1 | 1 | 1 | 1 | 124.40 | 125.57 | 18 | 1 | −1 | 1 | 109.33 | 111.30 |
| 2 | 1 | 1 | −1 | 106.14 | 98.34 | 19 | −1 | 0 | 0 | 127.49 | 120.10 |
| 3 | 0 | 0 | 0 | 103.36 | 114.67 | 20 | −1 | −1 | 1 | 116.73 | 114.44 |
| 4 | 0 | 0 | 0 | 106.91 | 114.67 | 21 | −1 | 1 | 1 | 142.92 | 143.15 |
| 5 | 0 | −1 | 0 | 101.42 | 99.36 | 22 | −1 | −1 | 1 | 112.54 | 114.44 |
| 6 | 0 | 0 | 1 | 125.28 | 127.86 | 23 | 1 | 1 | −1 | 98.32 | 98.34 |
| 7 | 1 | −1 | −1 | 77.95 | 77.21 | 24 | −1 | 1 | −1 | 110.48 | 102.05 |
| 8 | 0 | 1 | 0 | 124.38 | 126.11 | 25 | 0 | 0 | 0 | 122.5 | 114.67 |
| 9 | −1 | −1 | −1 | 66.49 | 67.81 | 26 | 0 | −1 | 0 | 97.76 | 99.36 |
| 10 | −1 | 1 | 1 | 142.06 | 143.15 | 27 | 1 | 0 | 0 | 114.32 | 118.81 |
| 11 | 0 | 0 | 1 | 132.47 | 127.85 | 28 | 1 | −1 | −1 | 76.49 | 77.21 |
| 12 | 0 | 0 | 0 | 121.50 | 114.67 | 29 | 1 | 0 | 0 | 123.3 | 118.81 |
| 13 | 0 | 0 | −1 | 82.57 | 82.64 | 30 | 0 | 1 | 0 | 127.85 | 126.11 |
| 14 | 0 | 0 | −1 | 86.80 | 82.64 | 31 | 1 | 1 | 1 | 125.49 | 125.57 |
| 15 | 1 | −1 | 1 | 111.37 | 111.30 | 32 | −1 | 1 | −1 | 114.83 | 102.05 |
| 16 | 0 | 0 | 0 | 120.50 | 114.67 | 33 | −1 | 0 | 0 | 120.16 | 120.10 |
| 17 | −1 | −1 | −1 | 69.05 | 67.814 | 34 | 0 | 0 | 0 | 120.9 | 114.67 |

Following the testing process, the actual (measured) values were compared with the predicted values obtained from the ANN analysis. The models offered the best prediction values by inserting them in the calculations of the $R^2$ (Equation (3)), RMSE (Equation (4)), MAE (Equation (5)) and SSE (Equation (6)) that are well-known useful performance functions.

$$R^2 = \left[ \sum_{i=1}^{n} \left( x_i - \bar{x} \right) \left( y_i - \bar{y} \right) / \sqrt{ \sum_{i=1}^{n} \left( x_i - \bar{x} \right)^2 \sum_{i=1}^{n} \left( y_i - \bar{y} \right)^2 } \right]^2 \tag{3}$$

$$RMSE = \sqrt{ \frac{1}{n} \sum_{i=1}^{n} (x_i - y_i)^2 } \tag{4}$$

$$MAE = \frac{ \sum \left| x_i - \bar{x} \right| }{ n } \tag{5}$$

$$SSE = \sum_{i=1}^{n} \left( x_i - \bar{x} \right)^2 \tag{6}$$

where $n$ is the number of observations, $x_i$ and $\bar{x}$ are the observed and their average values, $y_i$ and $\bar{y}$ are the related predicted and average values, respectively.

As $R^2$ increases and approaches 1, the predicted values approach the experimental values, showing the high ability of the model to predict the response with a high precision. As the errors become minimum, the model being examined also offers the response prediction more precisely.

The structure of the prediction model network including one input layer, one hidden layer and one output layer. In the ANN structure, the F to MU molar ratio (MR), the weight ratio of the modified protein to the MUF resin (WR) and the press temperature (Tem) were chosen as the input variables while the MOR was chosen as the output variable. The number of the process elements (neurons) of the hidden layer has been 6 for the model.

To determine the MOR prediction model, the feed forward backpropagation multi-layer ANNs were used. In the model offered, the hyperbolic tangent sigmoid function (*tansig*) was preferred as the transfer function in the hidden layer while the linear transfer function (*purelin*) was used in the output layer. After testing the errors obtained using Levenberg-Marquardt backpropagation (*trainlm*), scaled conjugate gradient (*trainscg*) and Bayesian regularization backpropagation (*trainbr*) algorithms as the training algorithms and determining the suitable algorithm with the least error, the momentum gradient reduction backpropagation algorithm (*trainlm*) was used to train the rules, and the mean square error (MSE) calculated by the Equation (7) was preferred as the performance function.

$$MSE = \frac{1}{N}\sum_{i=1}^{N}(t_i - td_i)^2 \tag{7}$$

where *ti* is the actual output (target values), $td_i$ is the neural network output (estimated values) and *N* is the number of training patterns.

To involve the models for each parameter uniformly, data of the training, testing and validation data sets were normalized from −1 to +1 when the hyperbolic tangent sigmoid function was used in the models and then, data were de-normalized to their original values so that the results were interpretable. The normalization operation was possible by applying the Equation (8):

$$X_{nor} = 2 \times \frac{X - X_{min}}{X_{max} - X_{min}} - 1 \tag{8}$$

where $X_{nor}$ is the normalized value of the variable *X* (the actual value of the variable) and $X_{max}$ and $X_{min}$ are the maximum and minimum values of *X* respectively.

### 2.3. Characterization Analysis

To examine the changes in the surface functional chemical groups, the Fourier transform infrared (FT-IR) spectroscopy analysis was performed using the pelletized samples. About 100 mg of potassium bromide (KBr) was mixed with 2 mg of the ground sample of the cured index adhesives. The prepared samples were scanned using the Thermo Scientific Nicolet 6700 FT-IR Spectrometer (Thermo Fisher Scientific, Waltham, MA, USA) in the wavelength range from 600 to 4000 cm$^{-1}$.

The thermogravimetry analysis was performed to receive the thermal behavior of the index adhesives. For this purpose, Shimadzu TGA-50 apparatus (Shimadzu Corporation, Kyoto, Japan) was used applying 10 mg of the powdered sample. The samples were heated from 30 to 400 °C with the heating speed 10 °C/min with the nitrogen flow fixed at 20 mL/min to create a neutral environment.

### 3. Results and Discussion

The model was trained, validated and tested using 34 data (Table 2). Figure 2 shows the error changes of the chosen neural network graphically. It is observed that the performance function reached its minimum value (16.2269) at the end of the second iteration (epochs) for the MOR.

The regression analysis is used often between the predicted and measured values to validate the networks. The accuracy of the models' estimation increases when the coefficients of correlation or determination go to one. It shows that there is a perfect fit between the actual and predicted values here. Figure 3 shows the correlation between the calculated and actual values (training $R^2$ = 0.964, testing $R^2$ = 0.9111). $R^2$ in the testing set shows that the network explains at least 91.11% of the actual data. These values prove that the developed models have good performance and support the application of the ANNs predictors. The normality hypothesis evaluates through the x-y normal plot that the defined points must follow a linear form if the data values are the result of a normal distribution. x-y plot indicates that the residuals are distributed almost normally. However,

the plots did not deviate from the expected identity line. These results confirm that the data support the ANN hypotheses.

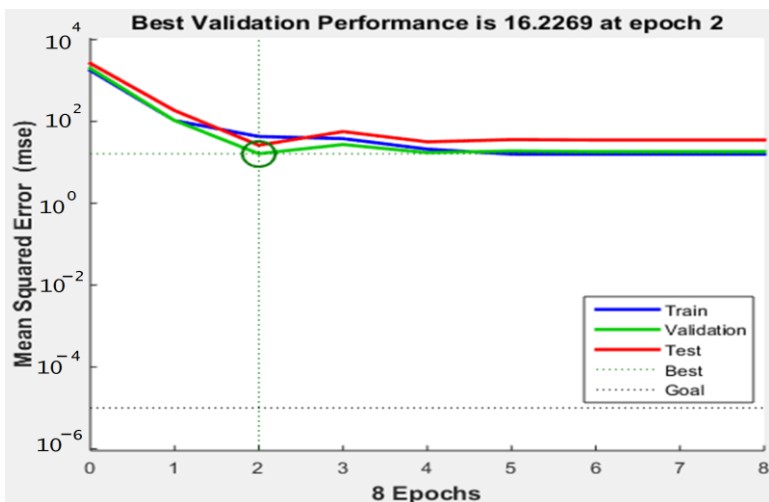

**Figure 2.** Changes in the MSE at each iteration for the MOR estimation models.

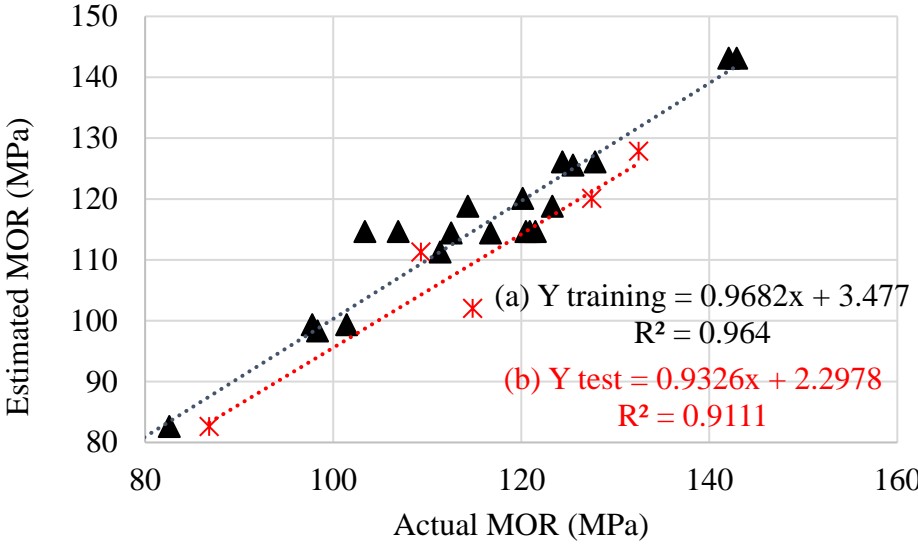

**Figure 3.** The comparison of the estimated and actual values (a) training data set, (b) testing data set.

As it was stated before, RMSE, MAE and SSE were employed to evaluate the model's performance. The RMSE, MAE and SSE results (Table 3) indicate that the trainlm algorithm has the least error in the training, testing and validation data sets and also in all data sets with the RMSE values including 3.944, 7.218, 15.305 and 4.962, MAE values including 2.663, 6.184, 5.563 and 2.607 and SSE values including 375, 260, 201 and 837 compared to the trainscg and trainbr based on the RMSE, MAE and SSE statistics. Hence, the optimum network structure was obtained with the minimum RMSE, MAE and SSE values to estimate the MOR in a network with 6 neurons in the hidden layer (1-6-3). The low RMSE value is a parameter that indicates whether the performance of the chosen model is suitable or not [39]. It is observed in Table 3 that it is better to evaluate the model based on the testing set due to more data points in the training set that can lead to more likely high non-normal values [40]. In sum, it was determined that the whole performance of the model decreased as the number of neurons increased beyond a set limit. Similar results were also obtained by Schaop et al. [41] that stated that ANN is more practical and reliable if the number of neurons is set in terms of the number of inputs.

**Table 3.** The comparison of the ANN models generated for the MOR estimation.

| Source | TrainLM | | | | TrainSCG | | | | TrainBR | | | |
|---|---|---|---|---|---|---|---|---|---|---|---|---|
| | Tra. | Tes. | Val. | All | Tra. | Tes. | Val. | All | Tra. | Tes. | Val. | All |
| RMSE | 3.954 | 7.218 | 15.305 | 4.962 | 10.138 | 13.759 | 19.996 | 19.996 | 4.857 | 8.369 | 28.056 | 5.345 |
| MAE | 2.663 | 6.184 | 5.563 | 2.607 | 8.228 | 13.267 | 7.336 | 8.838 | 3.863 | 7.625 | 2.473 | 4.211 |
| SSE | 375 | 260 | 201 | 837 | 2466 | 946 | 440 | 3853 | 566 | 350 | 55 | 971 |

The residuals of the model were tested to check the variance linearity and homogeneity hypotheses. In Figure 4, the standardized residuals are scattered somewhat similarly above and below their zero middle, showing that the data have supported the ANN hypotheses. It is observed that more than 70% of the errors in the range below 5% and −3% are in the ANN model developed based on the trainlm algorithm.

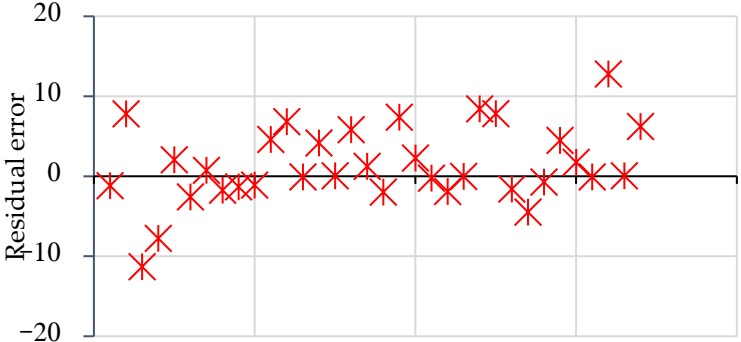

**Figure 4.** The residual error values for all data set.

The plots of the comparison between the actual and predicted values are given in Figure 5. The comparison of the actual and predicted values for the data sets not only showed the predictability of ANN to know the MOR data but it also proved the generalizability of the model for indefinite unknown MOR data. As it is observed, the values are very close to each other and there is a fit almost in all testing, training and validation data sets. It increases the applicability of the ANN model.

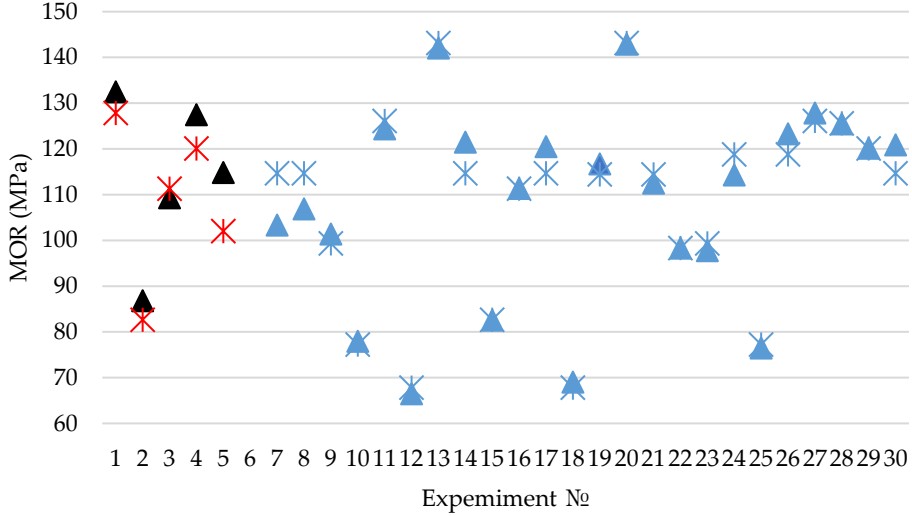

**Figure 5.** The comparison of the measured and predicted MOR values with the best-fit ANN model for the testing (black and red markers), training and validation (blue markers) data sets (triangle and star markers are actual and estimated values, respectively).

Researchers have used the experimental data to study the effect of F to U molar ratios, bio-based adhesive application, press time and temperature, etc on the bending strength through ANOVA or the regression tree method directly. Although ANOVA and the regression tree method based on mathematical analysis could provide more detailed information compared to 2D or 3D charts, the size of the data resulting from experiment could be limited normally. The precision of the analysis results must be affected based on several limited data. A reliable ANN bending strength prediction model can describe the mathematical relation between different parameters of making a composite and MOR manually. However, the ANN model was a black box through which the visible effects of the composite making parameters on the MOR could be offered hardly. Using the ANN, it is possible to produce enough data to analyze the making parameters affecting the MOR through 2D and 3D charts. For instance, if one is interested in the effect of WR on MOR, data can be produced that are given in Table 4. Then, the curve analysis can be obtained like what is presented in Figure 6. If one is interested in studying the interactive effects of MR × WR, MR × Tem or WR × Tem on the MOR, data can be produced like what is given in Table 5. Then, the curve analysis can be obtained like what is offered in Figures 7 and 8.

**Table 4.** Example data for analyzing a single parameter's influence on MOR.

| No. | MR | WR | Tem (°C) | MOR (MPa) |
|-----|------|----|----------|-----------|
| 1 | 1.68 | 20 | 160 | 91.203 |
| 2 | 1.68 | 25 | 160 | 99.358 |
| 3 | 1.68 | 30 | 160 | 107.51 |
| . | . | . | . | . |
| . | . | . | . | . |
| . | . | . | . | . |
| 8 | 1.68 | 55 | 160 | 126.64 |
| 9 | 1.68 | 60 | 160 | 127.57 |

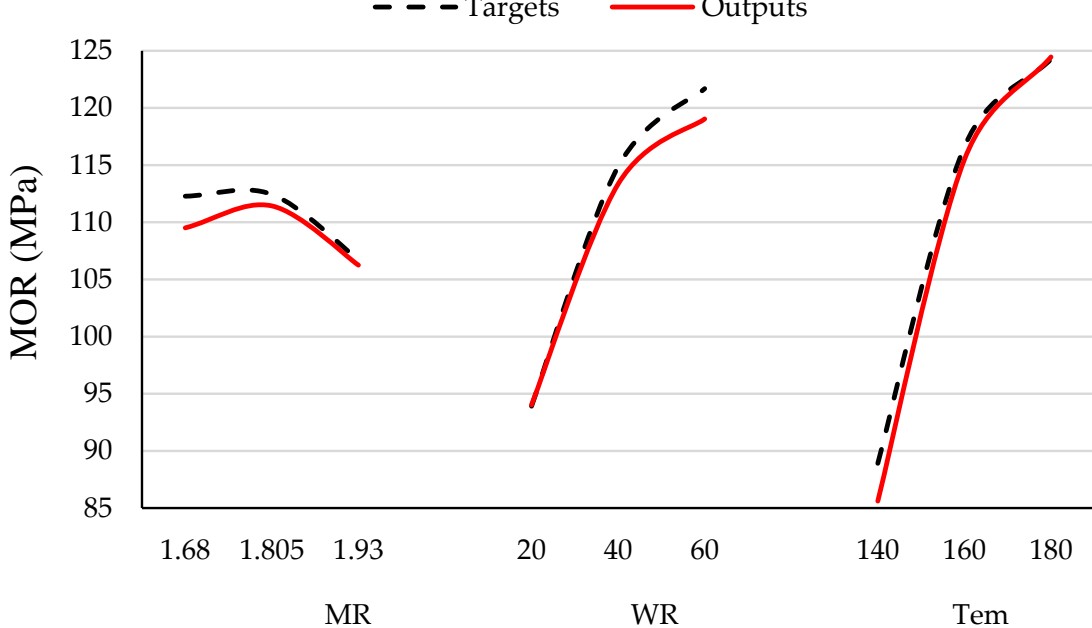

**Figure 6.** The direct effect of parameters on MOR and comparison of them with the best-fit ANN model.

**Table 5.** Example data for analyzing multiple parameters' influence on MOR.

| No. | MR | WR | Tem (°C) | MOR (MPa) |
|---|---|---|---|---|
| 1 | 1.68 | 20 | 140 | 91.203 |
| 2 | 1.71125 | 25 | 140 | 101.46 |
| 3 | 1.7425 | 30 | 140 | 111.71 |
| . | . | . | . | . |
| . | . | . | . | . |
| . | . | . | . | . |
| 82 | 1.68 | 20 | 140 | 90.12 |
| 83 | 1.71125 | 20 | 145 | 97.234 |
| 84 | 1.7425 | 20 | 150 | 104.26 |
| . | . | . | . | . |
| . | . | . | . | . |
| . | . | . | . | . |
| 243 | 1.93 | 60 | 180 | 134.36 |

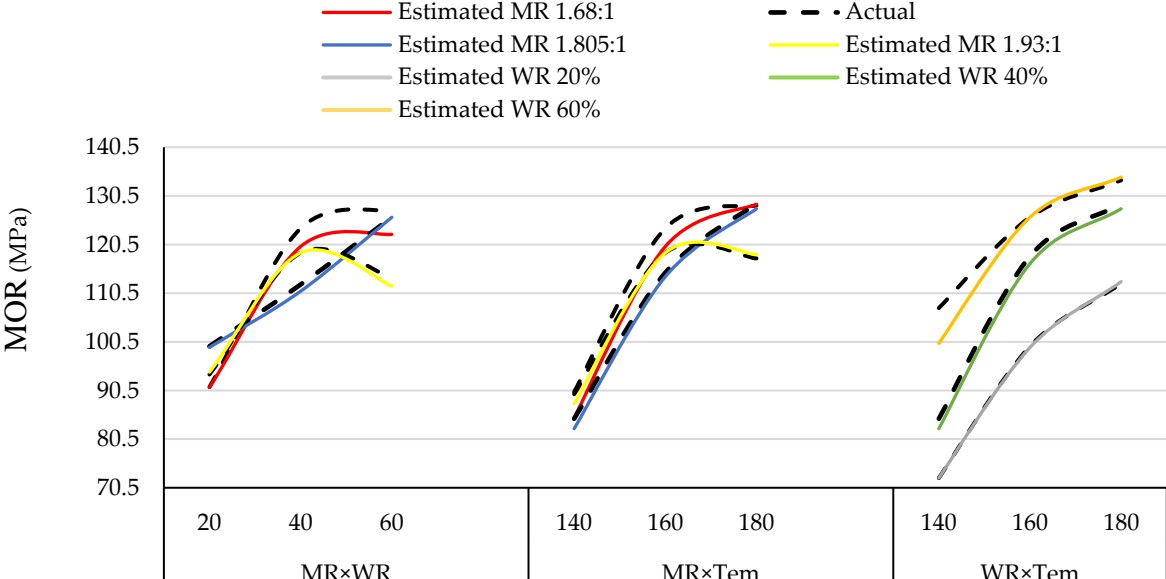

**Figure 7.** The interaction effect of parameters on MOR and comparison of them with the best-fit ANN model.

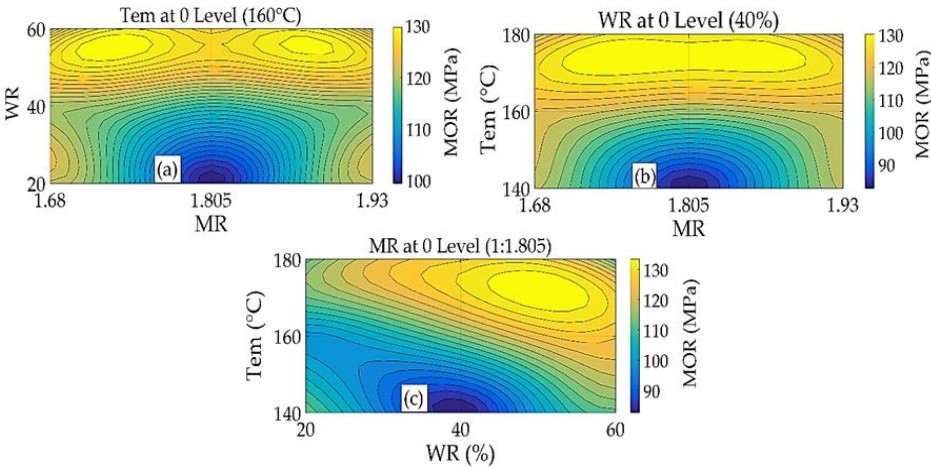

**Figure 8.** Counter plot of interaction effect of MR × WR (**a**), MR × Tem (**b**), and WR × Tem (**c**) on MOR estimated by ANN.

Similarly, the curves showing the direct effect of MR, WR and Tem on MOR are given in Figure 6 and the interactive effect of the independent variables on the MOR is given in Figures 7 and 8. It is evident in Figure 6 that the average MOR increases slightly as the MR increases and it decreases sharply afterwards. The changes in the actual MOR are largely similar to those of the estimated MOR so that there is a perfect match at the maximum MR. As the protein content increases to 40%, the increase in MOR is linear, and the intensity of its increase decreases afterwards and becomes nonlinear. The maximum MOR is where the MUF resin is replaced by the maximum protein. The changes in the actual MOR agree largely with the estimated value while the error between both values increases slightly as maximum protein is used. As the press temperature increases to the middle level (160 °C), the actual and estimated MOR increase linearly while as the press temperature increases to 180 °C, the changes in the increase of MOR increase nonlinearly (125 MPa). There is a perfect match between the actual and estimated values of MOR in the effect of the changes in the press temperature.

It is observed in Figure 7 that there is a perfect match between the estimated values of MOR resulting from the effect of MR × WR (x1x2), MR × Tem (x1x3) and WR × Tem (x2x3) when the third factor of each group of the interactive effects is at the middle level. Different statistics being studied such as R2, RMSE, MAE and SSE also confirm the agreement or in other words, the minimum error between the estimated and actual values (Table 3). According to the nonlinear regression model resulting from the MOR prediction (Equation (9), y ~ f(b,x)), the regression coefficients of the direct, interactive and square effects of the variables together with the related mathematical signs (− or +) indicate that MOR is affected differently for each source of change when the sources have been significant statistically. It is observed that according to the significance of each source of change, the direct effect of the change in the press temperature (19.427) results in the maximum change in the response being examined followed by the direct effect of the protein content (12.51). However, the square effect of Tem (−10.648) has the maximum effect on the MOR decrease followed by the interactive effect of MR × WR.

$$y \sim f(b,x) = 115.37 - 1.632x1 + 12.51x2 + 19.427x3 + 3.5542x1^2 - 3.1611x2^2 - 10.648x3^2 - 3.4452x1x2 - 3.3005x1x3 - 1.5461x2x3 \tag{9}$$

The trained ANN can provide the intermediate values required for optimizing the production process. In other words, thanks to the well-trained model, the outputs can be detected and tracked according to certain input values with a high precision without doing more experimental studies [42]. All outputs of the effects of the parameters on the dependent variable can be predicted by ANN for various combinations. The MOR values estimated by the ANN prediction model are given in Figures 7 and 8 according to different levels of MR, the WR and Tem. When the Tem is fixed at the middle level (160 °C), the prediction of the interactive effect of MR and WR on MOR indicated that as WR increases for each MR (ranging from minimum (1.68) to maximum (1.93)), MOR becomes maximum. As WR becomes minimum continuously, MOR approaches the minimum value and the minimum value is where MR is at the middle level (1.805) (Figure 8a). Also, in other trial, while WR was fixed at the middle level (40:60) and MR and the Tem changed, the results indicated that as the Tem increases continuously from 140 to 180 °C for all ranges of MR from 1.68 to 1.93, MOR becomes maximum. However, the minimum MOR decreases continuously as the Tem decreases and its minimum value is where MR is at the middle level (1.805) (Figure 8b). The relationship between MOR and the interactive effect of WR and Tem is shown in Figure 8c. It is observed that the interactive effects of both parameters are less than the interactive effects of other parameters on MOR. While MR is at the middle level (1.805), as WR increases to maximum and Tem becomes maximum, MOR also becomes maximum. However, as MR decreases to minimum and Tem becomes minimum simultaneously, MOR becomes minimum. By increasing the molar ratio of formaldehyde to melamine urea, more di hydroxymethylureas and trihydroxymethylureas are produced in the MUF resin and it causes the molecular weight to be higher with longer chains resulting

from methylene and methylene ether bonds and more branched chains [43]. On the other hand, by using $NH_4Cl$, the coagulation speed and temperature can be reduced as a result of the release of hydrochloric acid, so that the higher the molar ratio of formaldehyde, the higher the coagulation speed of MUF [44]. But due to the non-use of $NH_4Cl$, which was due to the possibility of determining the effect of hydrolyzed protein on the coagulation speed of MUF resin, increasing the temperature has completed the polycondensation reaction.

The viscosity of UF molecules is constant for all ranges of the shear rate due to their small size showing Newtonian behavior. However, protein or protein compound shows a non-Newtonian behavior even for its small amount mixed with UF resin and can align with the shear flow in contrast to the UF resin [45]. As a result, producing the shear thinning behavior, the modified protein can act as a rheological modifier in the UF resin. The result of this behavior is the increase in the apparent viscosity for low amounts of protein addition due to the interactive effects of protein-protein and protein-water through hydrogen bonds [46]. However, as the protein content increases more, viscosity decreases at high shear ranges ($\approx 103$ s$^{-1}$) until it reaches the amounts similar to the UF amounts and a rheological behavior is generated similar to when the adhesive (UF resin) is applied [47]. In this process, even as protein increases slightly, an entangled structure with a pseudoplastic behavior forms [47], showing a gel-like behavior. The result of this behavior is the increase in the suspension structure strength and the glue line building stability in which if protein is treated by NaOH, the occurred denaturation increases the polymer's random coil volume [48] so that more entanglement occurs in the adhesive emulsion containing more modified protein. Since the UF resins have three distinct variation stages as a function of the shear rate including shear thinning—Newtonian—shear thinning behaviors respectively at the beginning, middle and end of a viscosity-shear rate curve while the combination of the UF and modified protein only contains two regions of shear-thinning and approximate Newtonian behaviors respectively at the beginning and end of such a curve [6], protein molecules can be considered as an effective rheological modifier for MUF. In this way, protein plays the role of a filler or additive. Meanwhile, according to one of the main methods of adhesive application (roller coating), as protein increases, the adhesive experiences a deeper shear thinning behavior so that it can be an advantage for the roller coating method. However, due to the lack of proper activation of spherical particles rich in unmodified protein, the possibility of proper connection or good support with UF resin molecules and as a result its deposition in the suspension is not provided, so that three distinct points of shear thickening—shear thinning—approximate Newtonian behaviors appear in the viscosity-shear rate curve again [6]. Hence, according to the protein activation level and the intensity of its treatment or the increase in its amount in the adhesive, the bonding strength will increase continuously.

Due to the presence of molecules with a high molecular weight, the addition of the modified protein presents a complete pseudoplastic behavior so that the entanglement between polymer chains will increase. The increase is apparently related to the effect of the chemical crosslinkings inside the adhesive bulk and the chemical reactions between the adhesive and substrate (wood) in the curing stage. The chemical cross-linking created between the functional groups of carboxyl and uncoiled amine of peptides with active sites of the MUF resin molecules in the curing stage leads to the formation of a hard 3D network with a high molecular weight of the polymer connected through covalence linkages [49,50]. This phenomenon increases the cohesive strength inside the adhesive bulk while the entanglements of the polymer chains increase simultaneously that protect them from creeping during the mechanical bending test [51]. Hence, the chemical cross-linking between the components of the adhesive system and the potential chemical interactive effects between the adhesive and adherened in the curing step leads to the increase in the strength, not only in the bending but also during the horizontal shear that can result in delamination.

While a very high depth or very low depth of adhesive penetration leads to the decrease in the bonding strength [52,53], the optimum penetration with a certain degree can

increase the strength of the adhesives containing protein by developing 3D regions in the interphase region. Adhesives with polymers with a very high molecular weight or viscosity have a low depth penetration that result in ineffective mechanical interlocking and decrease the bonding strength [52–54]. Properties such as viscosity and penetrability of the uncured adhesive are affected by the intermolecular interactive effects and the extent of polymer crosslinking [55]. Based on these proved facts, the press pattern at higher temperature can develop the intermolecular interactive effects of peptides and resin molecules through two factors: (1) high interactive effects between polymers lead to the increase in the molecular weight and decrease in the penetrability of the cured adhesive and (2) high intermolecular interactive effects lead to higher crosslinking in the curing step through which there is only a lower number of active groups accessible to form connection with wood in the curing step [50]. In these conditions, it was observed during the test that the failure mode is due to a middle level rupture of wood and interphase rupture in practice. It means that in effect using more protein along with a higher press temperature, the adhesion strength has reached the cohesion strength presented by wood and adhesive.

In the bending test, it was observed practically that as the MUF to protein ratio increased, horizontal failure occurred between the wood layers along the glue line (delamination). However, as the protein weight ratio increased, failure occurred under pure bending mode. Due to using no hardener such as $NH_4Cl$ that accelerates the self-curing reaction of the MUF resin during the hot press and forms some $CH_2$-O-$CH_2$ bridges that are weaker than methylene bonds, chemical bonds can form that are composed of protein-CH2-MUF bridges during the hot press. The necessity of using the MUF resin combined with the soy adhesive to the extent 40% is reported [56].

With a structure containing hydrophilic regions covered by hydrophobic regions and preventing the polar groups access, protein has a conformation including disulphide bonds, non-covalent forces such as van der waals interactive effects, hydrogen bonds and electrostatic interactive effects. During its treatment by NaOH, the tertiary and quaternary structures are destroyed to some extent and the functional groups are exposed due to the failure of the chemical bonds and intermolecular interactive effects of bonds and crosslinking [57]. NH2 of the hydrolyzed protein reacts with the free formaldehyde resulting from the absence of methylene connection that could occur between urea and formaldehyde and meanwhile, the higher the content of the hydrolyzed protein or the intensity of protein hydrolysis is, the higher its reaction with formaldehyde due to the exposure of more functional groups on protein will be.

Increasing protein and press temperature when F to M/U molar ratio is minimum, the strength becomes maximum. It means that at the presence of more functional groups of modified protein and higher press temperature, crosslinking occurs between these groups and NH2. However, if formaldehyde molar ratio increases before protein reacts with NH2, NH2 group of urea will have a methylation reaction with formaldehyde while the functional group of protein has not reacted with NH2 and no curing has occurred yet [57]. In these conditions, the modified protein can have a copolymerization reaction with urea/melamine and formaldehyde and form a network structure. This increase can be due to the lower remaining moisture along with the application of a higher range of temperature.

*Characterization Analysis of Results*

The FTIR spectrum of different adhesives is presented in Figure 9. The observed peak at about 3340 cm$^{-1}$ is related to the free and connected OH groups and N-H bending vibrations that can form hydrogen bonds with carbonyl group and peptide bonds in protein and the wood surface. It is observed that as WR increased, the peak has become wider with a lower intensity. However, as the press temperature increased, the peak's intensity has decreased again, showing the possibility of hydroxyl groups removal due to making new connections with protein. The observed peak at about 2906 cm$^{-1}$ is due to the symmetric and asymmetric stretching vibrations of methylene group in different adhesives. It is observed that the peak's intensity has increased in the adhesive containing more protein

content. However, as the curing temperature increased, the peak's intensity has decreased, showing that using a higher temperature, the modified protein could take part in the system's reaction successfully.

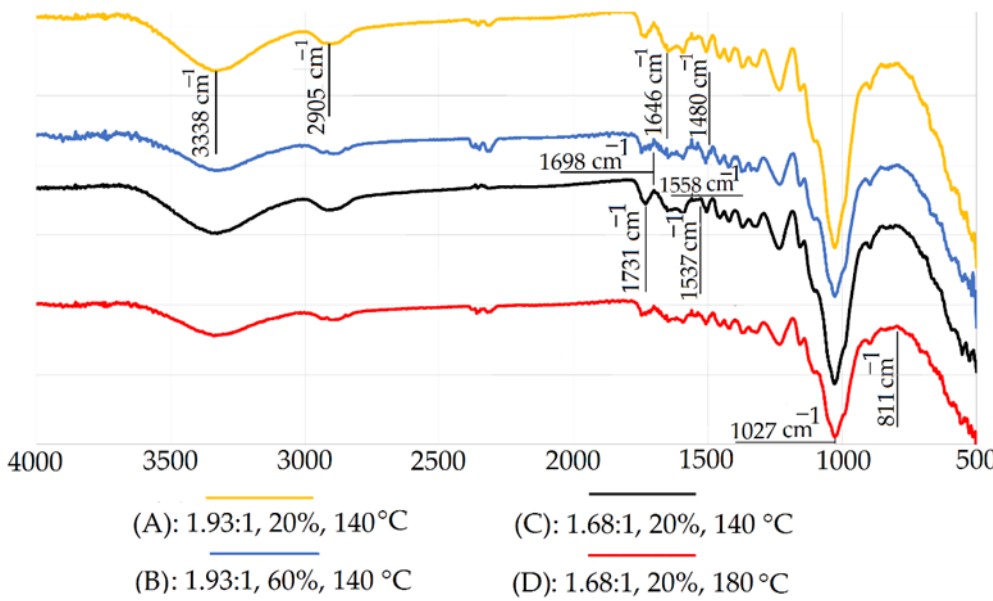

**Figure 9.** FTIR curves of adhesive containing different weight ratio of modified protein to MUF resin with different ratio cured at minimum and maximum press temperature.

Due to the presence of abundant flavonoid compounds in walnut wood, esterification of tannin hydroxyls by protein chain acids has resulted in the peak of ester bond at 1731 cm$^{-1}$. In this process, the reaction of protein with tannin flavonoid units can occur either through the amine group of protein's lateral chain or esterification with the acid group of protein's lateral chain [58]. It is observed that the widening and decrease in the intensity in this bond have occurred due to the increase in WR and Tem to some extent while MR had no effect. This means that even during the use of lower amounts of formaldehyde to melamine- urea, with the increase in protein consumption, probably the ester linkages tend to be replaced by methylene linkages that give maximum resistance to glue line of panel. Primary amide band (I type) in the range of 1630 to 1650 cm$^{-1}$ is characteristic of C=O stretching with a small amount of C-N stretching. As the modified protein increases mainly containing amine groups, imine bonds formation will shift from 1646 to 1698 cm$^{-1}$ offering the stretching bending imine (C=N). This bond shows the C=N formation in all amine compounds where formaldehyde reacts with amines to produce methyl- and ethyl-amine. It is confirmed by the increase in the peak's intensity at the band 1646 cm$^{-1}$ at the same time with the weakening of the peak 1698 cm$^{-1}$, showing that the adhesive has formed more stable chemical bonds and has formed a denser structure. It means that applying more modified protein, more cross-linking has formed with the MUF resin. The peak at 1558 cm$^{-1}$ shows the presence of N-C=N bending and ring deformation vibration in triazine ring [59]. As the protein content and press temperature increase, the obvious collapse in the peak and its development to 1537 cm$^{-1}$ indicate that the oxidized protein has entered the network structure in melamine's triazine ring effectively. Then, it may be accompanied by the deformation and widening of the peak concentrated at 3340 cm$^{-1}$ that is due to the complete overlap of different N-H and OH environments [60]. The peak at 811 cm$^{-1}$ also indicates triazine ring's out-of-plane vibration. This peak along with the peak at 1537 cm$^{-1}$ are very important characteristics of melamine's triazine ring [60]. The spectra are completely consistent with the expected structure in the MUF-MP adhesives. However, due to the decrease in the intensity of peaks at 811 cm$^{-1}$ in the samples containing more protein at a higher press temperature, active placement of protein can be considered as a permanent part of the adhesive, showing effective uniform distribution of protein in

the MUF matrix. Free amine groups of protein are able to react with formaldehyde and accompany the MUF resin structure cured by the application of the press temperature [56]. Based on the change in the intensity of the peak related to the C-O absorption of the modified adhesive at 1027 cm$^{-1}$ showing that C-O is affected by protein, it is evident that as the protein and press temperature increase, MUF is introduced to modified protein successfully due to the rapid reaction of MUFP hydroxymethyl active groups [61] and a lot of intermolecular hydrogen bonds are created with active reactive groups of the protein molecule such as amine [62]. A suitable agreement was obtained between the results obtained from FTIR analysis and the bending strength of the boards. It was found that by increasing the use of modified protein and increasing the temperature of the press, it was possible to create new connections between functional groups of protein with urea and melamine. During the optimization of the production process of layered products, the results indicated that the optimal value of bending strength is where the amount of protein used and the press temperature were at the maximum possible level, while the molar ratio of formaldehyde to melamine urea in it was moderate or low.

The thermogravimetric (TG) curve of the adhesive containing different MRs with different WRs cured at different Tems of panels is given in Figure 10, showing three certain stages of similar schematics of mass loss. These stages include water evaporation in the range of temperature less than 160 °C, decomposition of small molecules resulting from the modified protein broken by urea and melamine and also breakage of stable chemical bonds at the temperature between 160–320 °C due to the breakage of intra- and intermolecular hydrogen bonds, electrostatic bonds and division of covalent bonding between the remaining peptide bonds of amino acid [63] and destruction of S-S, O-N and O-O bonds and finally, backbone peptide bonds of protein in the adhesive at the temperatures above 320 °C producing gases such as CO, $CO_2$, $NH_3$ and $H_2S$ [64]. As the protein content increases in the compound, the weight-temperature slopes decrease. A similar trend is observed when MR, WR and press temperature increase (orange and black lines) and also when WR and press temperature are maximum but MR decreases (orange and red curves). These stages of mass loss are also reflected in relation with the derivative thermogravimetric (DTG) curve as shown in Figure 11. It is observed that as the press temperature increases continuously, DTV curve peaks decrease in the second stage (from black to orange lines). Simultaneously, as the modified protein added to the MUF increases, the intensity of the DTV curve peaks decreases in the second stage (red and blue line). In addition, as MR increases when the press temperature and the added protein content are maximum (blue and orange lines), DTV curve peaks become minimum in the second stage, showing that as the consumed protein becomes maximum, there is a maximum efficiency at the presence of high press temperature and more formaldehyde molar ratio. However, the difference between this treatment (orange line) and when minimum protein content and MR and maximum temperature are applied (blue line) is minimum. It means that at the presence of a lower protein content, application of a higher press temperature is more suitable. Meanwhile, the process of urea and melamine modification by protein has resulted in the formation of more stable chemical bonds and more smaller molecules, that have led to descending changes in the intensity of the DTV curve peaks, showing the formation of polysaccharide crosslinkings and dense network presenting a higher temperature stability. The XRD results showed that there is a perfect match between the changes in peaks resulting from increasing the amount of protein consumption and increasing the heat of the press and the bending strength values of the manufactured panels. Also, based on the change in the intensity of the peaks as a result of the change in the values of dependent variables, by reducing the amount of protein consumption, higher levels of the molar ratio of formaldehyde to MU can be used, while with the increase of protein consumption, the use of lower levels of the molar ratio of formaldehyde to MU is essential. These results are consistent with the optimized outputs by genetic algorithm coupled with ANN.

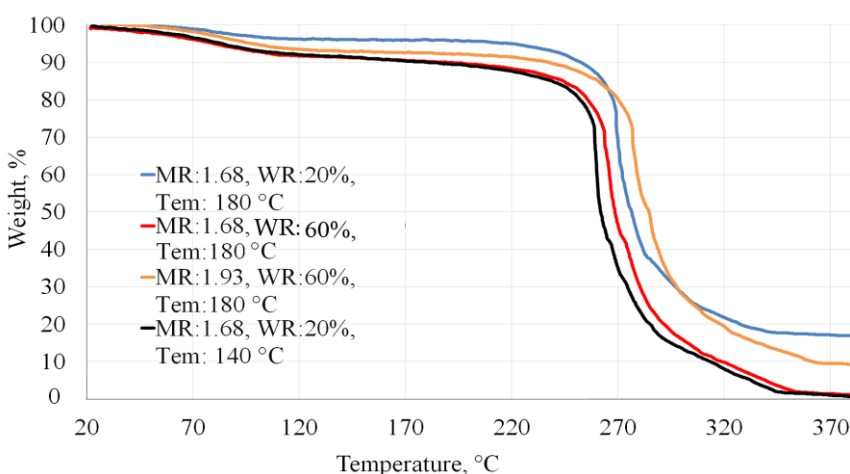

**Figure 10.** TG curves of adhesives.

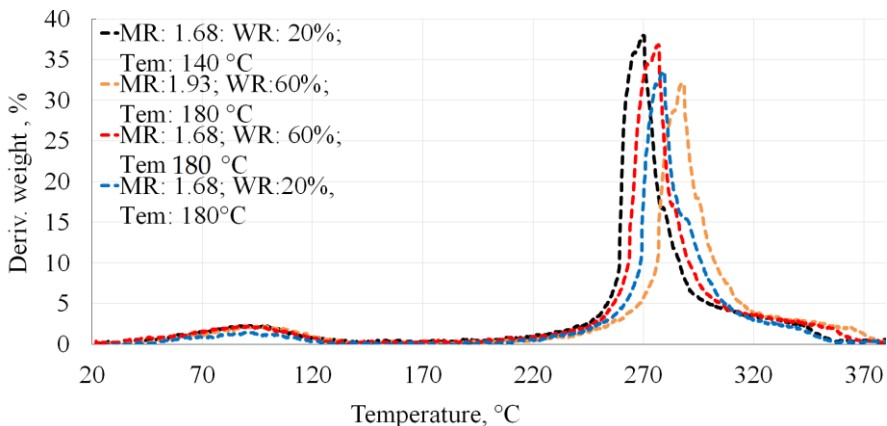

**Figure 11.** DTG curves of adhesives.

### 4. Conclusions

The present research has studied the ability of modeling the bending strength of the laminated products made of walnut wood adhered by the bio-based hydrolyzed soybean protein combined with MUF resin adhesive with different F to M/U molar ratios at different press temperatures. The results showed that:

- The bending strength changes significantly as the F to M/U molar ratio and the weight ratio of the modified protein to MUF resin change so that as F to M/U molar ratio decreases and the weight ratio of protein to MUF resin increases, the bending strength increases. Also, in the interactive effect of MR and press temperature, as the press temperature increases or decreases and the MR increases to a certain level and as MR approaches the maximum value, MOR decreases. Furthermore, in the interactive effect of the press temperature and weight ratio of protein to MUF resin, the increase in the temperature and WR will result in the increase in the bending strength.
- The evaluation between the experimental values and those predicted by ANN resulted in the presentation of an excellent relationship (with a difference less than 5%) for the estimated series of the process parameters.
- The ANN method could effectively produce experimental data resulting from the determination of the bending strength of the laminated wood products so that using suitable algorithms, ANN could offer a well-trained model to estimate the response being examined through which the experimental costs and time could be saved to determine the effect of each production variable on the response being examined.
- The diagnostic analysis presented by FTIR and TGA showed that urea, melamine and free formaldehyde in resin could interact chemically with the modified soy protein and

improve the bending strength of the laminated product so that as the modified protein increased compared to the MUF resin, the chemical interactive effects intensified along with the decrease in the F to M/U molar ratio.

**Author Contributions:** Conceptualization, methodology, software, validation, formal analysis, investigation and resources, project administration, writing and original draft preparation, M.N. and F.N.; investigation, visualization, writing—review and editing writing and supervision, A.N.P. All authors have read and agreed to the published version of the manuscript.

**Funding:** This research received no external funding.

**Data Availability Statement:** Not applicable.

**Conflicts of Interest:** The authors declare no conflict of interest.

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
