# Peer review of "Application of the Artificial Neural Network to Predict the Bending Strength of the Engineered Laminated Wood Produced Using the Hydrolyzed Soy Protein-Melamine Urea Formaldehyde Copolymer Adhesive"

_jcs, doi:10.3390/jcs7050206_

Round 1

Reviewer 1 Report

This manuscript by Nazerian et al. examined the bending strength of the laminated products adhered by the plant protein adhesive resulting from the soybean and melamine-urea formaldehyde (MUF) resin. The article has done a lot of work, but there are still some issues that need improvement as follows:

1. Too many miscellaneous peaks appear in the FT-IR spectra in Figure 9, and the typical peaks are not clearly observed. Please re-do the test.

2. The DTA and TGA analysis in Figure 10 should be separated.

3. The overall format of the text should be further adjusted according to the guide for authors. 

Therefore, it can be considered for publication in Journal of Composited Science after revision.

Reviewer 2 Report

The manuscript describes the application of the artificial neural network to predict the bending strength of the engineered laminated wood produced using the hydrolyzed soy protein-melamine urea formaldehyde copolymer adhesive.

The text is well-written, some points are raised below.

1.           2.2.2: Is the preparation novel? If not, please, add suitable references.

2.           Abstract: full name for MOR needed.

3.           How was the bending strength determined experimentally?

4.           P.12: “However, as MR decreases to minimum and Tem becomes minimum simultaneously, MOR becomes minimum”. Please, explain the theoretical basis for this observation.

5.           P.13: “The viscosity of UF molecules is constant for all ranges of the shear rate due to their small size showing Newtonian behavior.” Please, include the experimental data for the viscosity measurements. A relevant figure with the results would be useful.

6.           Ps.13-15: If there are no experimental data on the viscosity, these three pages should be shortened at least 50%.

7.           P.16: “However, MR had no effect, showing that as the protein content increases, ester bonds probably tend to be replaced by methylene bonds.” Please, clarify.

8.           P. 16: What is “I amide bond”?

9.           P. 17: “The peak at 811 cm-1 also indicates triazine ring's out-of-plane vibration. This peak along with the peak at 1537 cm-1 are very important characteristics of melamine's triazine ring.”. References are needed.

10.       A correlation between the FTIR data and the “the application of the artificial neural network to predict the bending strength” should be established.

11.       A correlation between the TG data and the “the application of the artificial neural network to predict the bending strength” should be, also, established.

Round 2

Reviewer 1 Report

This paper could be accepted in present form.

The quality of the English language is acceptable.

Reviewer 2 Report

No comments